# Optimal IP Current Controller Design Based on Small Signal Stability for THD Reduction of a High-Power-Density PFC Boost Converter

**Ahmed H. Okilly** , **Hojin Jeong** and **Jeihoon Baek** *

Electrical & Electronics and Communication Engineering Department, Koreatech University, Cheonan-si 31253, Korea; ahmed21490@koreatech.ac.kr (A.H.O.); husik5864@koreatech.ac.kr (H.J.)
* Correspondence: jhbaek@koreatech.ac.kr; Tel.: +82-41-560-1258

**Abstract:** This paper presents an optimal design for the inner current-control loop of the continuous current conduction mode (CCM) power factor correction (PFC) stage, which can be used as the front stage of the two-stage AC/DC telecom power supply. The conventional single-phase CCM-PFC boost converter is implemented with proportional–integral (PI) controllers in both the voltage and current-control loops to regulate the output DC voltage to the specified value and to ensure the input current follows the input voltage, which offers a converter with a high-power factor (PF) and low current total harmonic distortion (THD). However, due to the slow dynamic response of the PI controller at the zero-crossing point of the input supply current, the input current cannot fully follow the input voltage, which leads to high THD. In this paper, we investigate a digitally controlled PFC converter with an optimally designed inner current-control loop using a doubly-fed control loops integral-proportional (IP) controller to reduce the THD and to offer an input current with a unity PF. For the economic design of a digitally controlled PFC converter, two isolated AC and DC voltage sensors are designed for interfacing with the microcontroller unit (MCU). PSIM software as well as experimental prototype was used to test the converter performance using the proposed designed current controllers and isolated voltage sensors. We achieved a high-power-density, digitally controlled, telecom PFC stage with a power factor more than 99% and THD of about 5.50%.

**Keywords:** CCM-PFC; small signal model; IP controller; zero-crossing point; total harmonic distortion (THD); power factor (PF); isolated voltage sensors

## 1. Introduction

Conventional AC/DC rectifiers, which consist of a bridge rectifier and smoothing capacitors, can be used to supply power in telecom applications, but the circuit performance and fixed control parameters in these rectifiers lead to high total harmonic distortion (THD), high power losses, and reduced conversion efficiency, especially in high-power-density applications. So, the high-power factor and efficiency requirements in telecom applications limit the use of such conventional rectifiers [1]. Currently, to offer efficient high-power-density power supply, the active-controlled AC/DC converters, with high power density based on the boost converter technique, have been widely used to regulate the power factor (PF), reducing THD and the circuit power losses, and increasing the conversion efficiency [2,3].

The AC/DC power supply with two stages, as illustrated in Figure 1, is the optimal configuration to obtain high values for the input PF and power conversion efficiency. Two-stage active AC/DC telecom power supply consists of (1) the active power factor correction (PFC) stage to offer input current with high PF and (2) the DC/DC output converter stage, which is used to regulate the bus voltage of the PFC converter stage (320–410 V) to the distribution load level (45–63 V) for telecom applications [4].

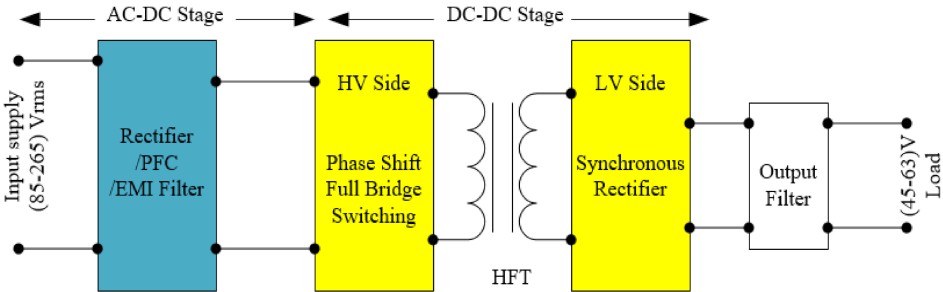

**Figure 1.** Block diagram of the two stages AC-DC telecom power supply.

According to the power applications, many types of PFC converters are widely used, such as the conventional [5–7], bridgeless [8,9], and the interleaved [10,11] PFC boost converters. For telecom applications, the conventional PFC boost converter is the most commonly used circuit because of its good performance, low cost, simple operations, and lower requirements for the power and control circuit designs [1,12]. The target of this study was to improve the performance of the conventional PFC stage telecom power supply by optimizing the control of the boost converter controllers to produce a converter with a unity power factor by reducing the THD of the input supply current.

A PFC boost converter controlling circuit can be implemented using digital or analog techniques. Controlling the circuit using analog integrated circuits (IC) offers an economic design of the converter; otherwise, it will have some disadvantages compared with the digital micro-controller unit (MCU), such as slow response, low operating temperature, and fixed control parameters. Digital MCU offers a high-speed response, flexible adjustment, and programming of the control techniques [13]; however, the price of the measurement sensors required for the interface between the converter power circuit and DSP MCU is high compared with sensors for interfacing with analog control IC.

The conventional PFC boost converter consists of power and control circuits, where the power circuit contains a bridge rectifier, energy storage inductor, single switch or multiple switches connected in parallel (to increase the converter current rating), fast switching diode, and an output bulk capacitor. The PFC boost converter control circuit usually implements two control loops: the inner current-control loop to control the inductor current, which makes the input current follow the input voltage and maintain the circuit PF at higher values, and the outer voltage loop, which maintains the output voltage value as the specified load value. Usually, these control loops are implemented using PI controllers.

For the output voltage control loop, the PI controller is enough to regulate the output voltage to the specified value [14], but for the inductor current-control loop, the slow response of the PI current controller at the current zero-crossing point distorts the inductor current, which leads to increasing the THD of the supply current [15]. Increasing the THD in the converter circuits causes higher power losses in all circuit components; further, it can lead to high current stresses and failures of the system insulation and protection [16]. Power quality and performance are required to maintain the THD value at the standard values, such as IEEE 519-20142 [17] and IEC 61000-3-2 [18], which suggests that to improve the circuit PF, by reducing the current stresses and power losses in the different parts of the system, THD should be kept at the lowest value within the given standard requirements [17,18].

The current THD in the digitally controlled PFC boost converters can be reduced using different techniques, such as passive harmonic filters connected in parallel with the input side. This technique can be used to reduce the THD for low switching frequency PFC converters; however, for high-frequency PFC converters where the electro-magnetic interface (EMI) filters connect on the input side of the converters, the reliability of the passive filters to decrease the THD is low due to the difficulty of tuning the filter's resonance frequency in the presence of the EMI filters [19]. Digital current filters, such as finite response impulse (FIR) and infinite impulse response (IIR) filters, can be optimally designed and used to moderate the feedback digital current signals for the inner current-control

loops to reduce the current error and thereby reduce the current distortion. Using the adaptive FIR filter to reduce the THD of the current conduction mode (CCM)-bridgeless PFC converter was proposed in [20].

Another widely used technique to reduce the THD in PFC converters is the variable on-time (VOT) switching control technique by optimizing the duty cycle of the PFC converter to decrease the zero-crossing distortion (ZCD) period [20–23]. The VOT switching control technique can reliably reduce the THD in PFC converters, but uses complex mathematical analysis of the converter operation around the zero-crossing point to optimize the exact dead time around the zero-crossing point for exact on-time switching prediction [24].

This paper presents the small-signal stability modeling of the conventional PFC boost converter. Based on the signal stability model, the optimal integral–proportional (IP) current controller consists of a double-loop control strategy, where the integral gain feed-forward returns the inductor current to the reference set point, and the proportional gain implemented in the feedback path increases the controller response. The IP controller is usually used in the control circuits of DC/DC converters and DC motor drive systems [25–27]. Compared with the PI controller, the IP controller has less overshoot and a fast-dynamic response [27]. So, the IP current controller with fast dynamic response was designed for the inductor current to track the reference current, removing the total distortion that occurs around the zero-crossing point and increasing the power factor. Two different current techniques based on the conventional PI controller and the proposed IP controller are presented in this paper; a comparative analysis was performed based on system stability and controller reliability to reduce the zero-crossing current distortion. For the economic design of the digitally controlled PFC converter, the isolated voltage sensors with inexpensive components for interfacing with the MCU analog–digital converter (ADC) was completely designed and is proposed here.

The remainder of this paper is organized as follows: Section 2 briefly explains the problem of high THD in the current zero-crossing point of PFC converters and the impact of this problem on the circuit PF. Section 3 outlines the operation principle and the complete design of the PFC boost converter employed. Section 4 presents the design technique for the proposed controlling circuits for the CCM-PFC converter based on small-signal stability modeling. Section 5 introduces the proposed isolated AC and DC voltage sensors for the economic and reliable operation of digitally controlled PFC converters. Section 6 provides the simulation of the complete designed converter with different control techniques. Section 7 provides the prototype design, the experimental verification and results of the complete designed PFC converter, Section 8 is the conclusion of the paper.

## 2. PFC Converter Total Harmonic Distortion (THD) and Power Factor (PF)

In a power system, the THD value should be kept as low as possible; a lower THD produces a higher power factor, higher efficiency, and lower current stress in power system components due to fewer peak currents. Usually, the THD value should follow the standard specifications, such as IEC61000-3-2 or IEEE 519-20142, for different classes of equipment. Ignoring the effect of current distortion, the input PF of the PFC converter can be expressed as

$$PF = \cos(\theta_{vs} - \theta_{is}), \tag{1}$$

where $\theta_{vs}$ is the voltage angle and $\theta_{is}$ is current angle for the input supply.

Equation (1) is not the full definition of the converter PF and usually, this mathematical expression is called the displacement factor (dF). The power factor calculation using Equation (1) can be applied only if the supply voltage and current waveforms are completely sinusoidal.

In most power electronic circuits, including the PFC boost converters, operation techniques and rapid changes between on and off states of these converters act as nonlinear loads, which can change the nature of the current so it is no longer a sinusoidal waveform. Figure 2 shows the supply current and voltages of the PFC boost converter, demonstrating the current distortion when the boost converter switch changes rapidly from on to off states.

This point is usually called the zero-crossing point. The conventional control techniques using PI controllers cannot fully remove the current distortion of the input current in the zero-crossing point due to the slow dynamic response of the current controller at this point [15], and the inductor current cannot track the reference current when the supply current changes rapidly from positive to negative values. The root mean square value of the input current ($I_{s\_rms}$) for the PFC converter can be expressed as

$$I_{s\_rms} = \sqrt{I_{dc}^2 + \sum_{n=1}^{\infty} I_{n\_rms}} \, , \tag{2}$$

where $I_{dc}$ is the average current component and n is the harmonic order. When n = 1, $I_{1\_rms}$ is the fundamental component of the supply input current. With an ideal voltage source that offers voltage only at the fundamental frequency ($Vs_{\_rms} = V_{1\_rms}$), the converter input average power ($P_{avg}$) can be calculated as

$$P_{avg} = V_{1\_rms} \times I_{1\_rms} \times dF \, . \tag{3}$$

Apparent power (S), which represents the supply active and reactive power components, still includes all of the current harmonics and can be expressed as

$$S = V_{s\,rms} \times I_{s\,rms} = V_{1\_rms} \times \sqrt{I_{dc}^2 + \sum_{n=1}^{\infty} I_{n\_rms}}. \tag{4}$$

From Equations (3) and (4), the full definition of PFC converter power factor that considers the current distortion can be expressed as

$$PF = \frac{I_{1\_rms}}{\sqrt{I_{dc}^2 + \sum_{n=1}^{\infty} I_{n\_rms}}} \times dF = DF \times dF \tag{5}$$

where DF represents the current distortion factor and is equal to $\frac{I_{1\_rms}}{I_{s\_rms}}$. The THD value related to the current distortion factor can be expressed as

$$DF = \sqrt{\frac{1}{1 + THD^2}} \, . \tag{6}$$

So, the power factor (PF) is the product of two factors called displacement factor (DF) and the distortion factor (DF), as given in Equation (5). The displacement factor (DF) depends on the phase shift between the voltage and current of the input supply, which is usually very low for AC/DC power converters (less than five degrees). The distortion factor (DF) depends on the total harmonic distortion of the current waveform.

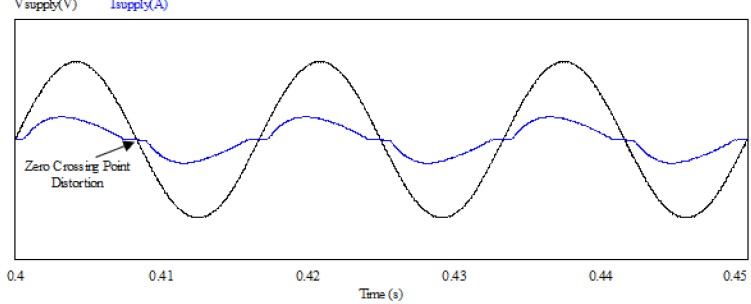

**Figure 2.** Supply voltage-current waveforms for PFC-boost converter with conventional PI current controller.

To clearly explain the relationship between the PF and THD in the PFC boost converters for a 5-degree phase shift between supply voltage and current, to achieve a PF of more than 99%, which is usually required in the specifications of telecom power applications [28], THD must be less than 11.20%.

### 3. Telecom AC/DC PFC Boost Converter Operation Principle and Design

Figure 3 shows the schematic circuit of the PFC boost converter, which usually consists of an electro-magnetic interface (EMI) filter connected to the AC input power source, bridge rectifier, and the boost converter unit, which is the main part of the PFC boost converter. The boost converter employed in this study works in current continuous conduction mode (CCM); this topology is the conventional and most used in telecom applications due to it is simple design, reliability for higher power applications, and low price compared with the other PFC topologies [1,12]. The boost converter unit consists of three main parts: the storage energy part, represented by the inductor ($L_b$); the switching elements, represented by high voltage switch ($Q_b$); high-speed switching diode ($D_b$); and the output filtering capacitor $C_b$.

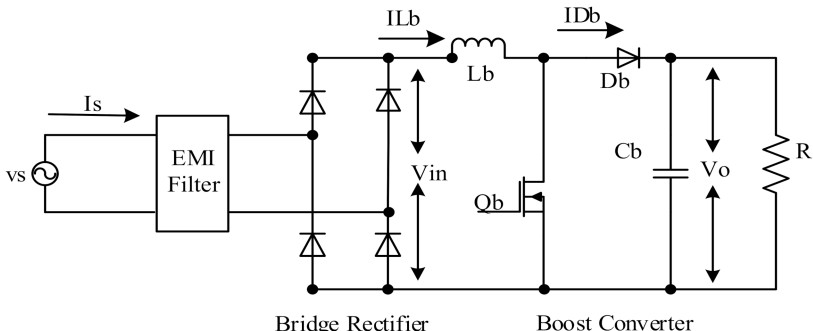

**Figure 3.** Schematic diagram of the conventional AC-DC PFC boost converter.

The following paragraphs present the operation principle and the design of the main components of the single-phase CCM-boost converter.

Figure 4 depicts the schematic circuits of the different operation modes for the CCM-boost converter. When the switch ($Q_b$) is closed, as shown in Figure 4a, the current flows through the energy storage element ($L_b$) in the direction marked by the red dotted line. At this moment, the energy is storage in the inductor, generating a magnetic field. When $Q_b$ is opened, as shown in Figure 4b, the current circuit impedance increases, thereby reducing the current, and the magnetic field previously created is reduced to maintain the current toward the load, as also marked by the red dotted line. Thus, the polarity of the inductor voltage is reversed, which places the two sources in series, providing a higher voltage to charge the capacitor ($C_b$) through the high-speed switching diode ($D_b$).

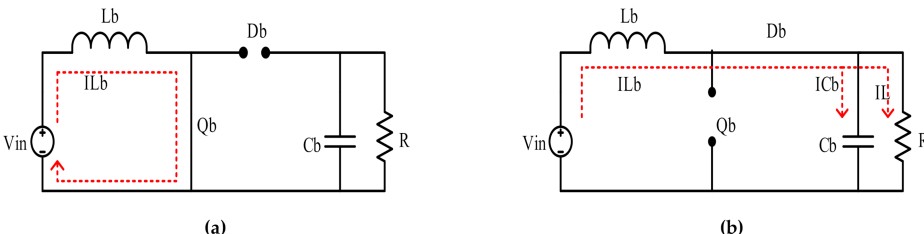

**Figure 4.** CCM-boost converter operation modes. (**a**) Switch $Q_b$ closed, (**b**) Switch $Q_b$ opened.

The three parts of the boost converter should be optimally designed to maintain the circuit-specified ripple currents, reduce the voltage stress in the switching elements during the converter operation, and maintain the output voltage with the ripple value required for telecom applications.

The PFC boost converter inductor ($L_b$) was designed based on the specified circuit ripple current (%Ripple) and the corresponding input and output voltage conditions [29]. PFC control circuits are usually designed to offer good performance for a wide range of input AC voltage (85–265 V). Using these design specifications, $L_b$ can be calculated as

$$L_b = \frac{1}{\%\text{Ripple}} \times \frac{1}{\eta} \times \frac{v_{s\ min}^2}{P_o} \left( 1 - \frac{\sqrt{2}\ v_{s\ min}}{V_o} \right) \frac{1}{\text{Fsw}}. \tag{7}$$

The inductor's maximum current ($I_{Lb\ Max}$) is determined based on the specified maximum current ripple [29] as expressed in

$$I_{Lb\ Max} = \frac{\sqrt{2}\,P_O}{v_{s\ min}} \times \left( 1 + \frac{\%\text{Ripple}}{2} \right). \tag{8}$$

where $V_{s\ min}$ is the minimum supply voltage's root mean square (RMS) value, $V_o$ is the rated output DC voltage, $\eta$ is the converter designed efficiency, $P_o$ is the output rated power, and $F_{sw}$ is the switching frequency.

The output filter capacitor, usually called the output bulk capacitor, was designed to meet the specified output voltage ripple using

$$C_o \geq \frac{P_o}{2 \times \pi \times F \times V_{rpp} \times V_o}. \tag{9}$$

where F is the input supply frequency and $V_{rpp}$ is the output voltage ripple peak-to-peak value. Additionally, the bulk output capacitor should be designed to offer the minimum voltage hold up with the specified time ($t_{hold}$), as expressed in

$$C_o \geq \frac{2 \times P_o \times t_{hold}}{V_o^2 - V_{o\ min}^2}. \tag{10}$$

The capacitor value was selected to have the larger value among the two equations.

Another important issue when choosing the output bulk capacitor is that the capacitor series equivalent resistance (ESR) should be very low, as it affects circuit efficiency and the output voltage regulations. A higher ESR causes more ripple, influencing the stability of the control loops [30]. Table 1 shows the designed specification and results for the 2500 W PFC boost converter employed in this work.

**Table 1.** Telecom CCM-PFC converter design specifications and components value.

| Parameter | Specification | Unit |
|---|---|---|
| AC Input voltage ($V_s$) | 220 (85–265) | V rms |
| AC Input frequency (F) | 60 (47–63) | Hz |
| DC output voltage ($V_o$) | 400 (320–410) | V |
| Rated power ($P_{out}$) | 2500 | W |
| Switching frequency ($F_{sw}$) | 100 | kHz |
| Input ripple current (%Ripple) | 10% | at full load |
| output ripple voltage ($V_{rpp}$) | 20 | V$_{-\text{peak to peak}}$ |
| Hold up time ($t_{hold}$) | 6 | ms |
| Output capacitor ($C_b$) | 1120 | µF |
| Boost converter inductor ($L_b$) | 470 | µH |

## 4. Design of the Proposed PFC Boost Converter Control Systems

A block diagram of the complete schematic circuit of the telecom CCM-PFC converter including the digitally controlled technique is shown in Figure 5. The CCM-PFC control circuit includes two control loops implemented inside the DSP MCU: the outer voltage control loop and the inner current-control loop (Figure 6). The outer loop is used for regulating the output voltage ($V_o$) of the PFC converter to the reference value and to

generate the voltage error signal, which is modified with the input sinusoidal voltage ($V_{in}$) to generate the reference current ($I_{ref}$) value for the inner control loop, as depicted in the control blocks in Figure 6. The inner loop is used to make the inductor current ($I_{Lb}$) track the refence current ($I_{ref}$); the inductor current is the DC value of the input supply current, so that any distortion in the inductor current will affect in the sinusoidal shape of the input supply current.

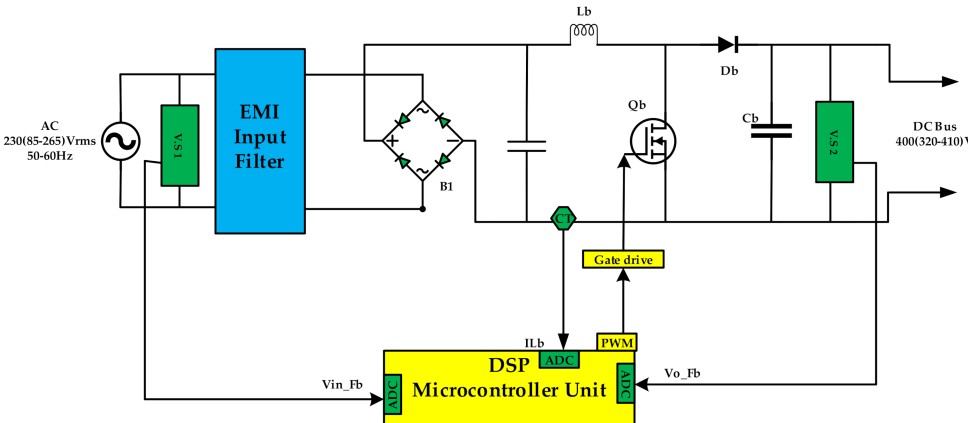

**Figure 5.** AC-DC telecom PFC boost converter with digitally controlled technique.

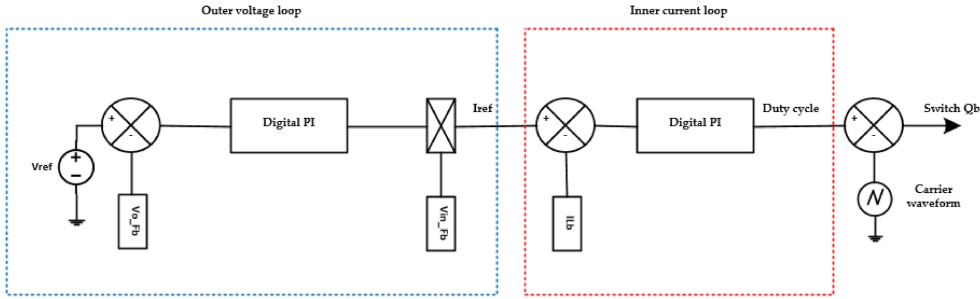

**Figure 6.** Implementation technique of the controlling circuit inside the DSP.

Usually, the PI controller is enough to regulate the output voltage to the reference limit, having good performance and fast response [14], so in the conventional and proposed control circuits, the same PI controller was used in the voltage control loop.

### 4.1. PFC Boost Converter Average Small-Signal Modeling

For the PFC boost converter shown in Figure 3, the inductor voltage ($V_{Lb}$) can be expressed as:

$$V_{Lb} = L_b \frac{dI_{Lb}}{dt} = V_{in} - V_o(1 - D). \tag{11}$$

The output capacitor current ($I_{Cb}$) can be expressed as

$$I_{Cb} = C_b \frac{dV_o}{dt} = -\frac{V_o}{R} + I_L(1 - D). \tag{12}$$

where D refers to the converter duty cycle, $V_o$ is the output bus voltage ($V_o = V_{bus}$), and R is the load resistance in ohms.

Assume all variables ($I_{Lb}$, $V_{in}$, $V_o$, and D ) are at the steady state at the selected operational point ($i_{Lb}$, $v_{in}$, $v_o$, and d ) and the small-signal AC variation ($i_{Lb}^*$, $v_{in}^*$, $v_o^*$, and $d^*$), where

$$I_{Lb} = i_{Lb} + i_{Lb}^* ; \; V_{in} = v_{in} + v_{in}^* ; \; V_o = v_o + v_o^* ; \; D = d + d^*. \tag{13}$$

The PFC boost converter control circuit shown in Figure 6 should modify the duty cycle (D) based on the small-signal AC variation to control the output voltage ($V_o$) and the inductor current ($I_{Lb}$).

Substituting Equation (13) into (11) and (12), we obtain

$$L_b \frac{d(i_{Lb} + i_{Lb}^*)}{dt} = (v_{in} + v_{in}^*) - (v_o + v_o^*)(1 - d - d^*), \tag{14}$$

$$C_b \frac{d(v_o + v_o^*)}{dt} = -\frac{(v_o + v_o^*)}{R} + I_{load}(1 - d - d^*). \tag{15}$$

Equations (14) and (15) can be re-written again as Equations (16) and (17) neglecting the term $v_o^* d^*$, which is the product of two AC small signals.

$$L_b \frac{d(i_{Lb}^*)}{dt} = (v_{in}^*) - (v_o^*)(1 - d) + v_o . d^* \tag{16}$$

$$C_b \frac{d(v_o^*)}{dt} = -\frac{v_o^*}{R} + I_{Lb} . d^* + (i_{Lb}^*)(1 - d) \tag{17}$$

Use the Laplace transform to obtain the small-signal model, and arrange the model in state space matrix form to obtain

$$\begin{bmatrix} sL_b & 1-d \\ 1-d & -sC_b - \frac{1}{R} \end{bmatrix} \begin{bmatrix} i_{Lb}^*(s) \\ v_o^*(s) \end{bmatrix} = \begin{bmatrix} v_o \\ i_{Lb} \end{bmatrix} . d^*(s) + \begin{bmatrix} 1 \\ 0 \end{bmatrix} . v_{in}^*(s). \tag{18}$$

With the derived model of the PFC boost converter and selecting the appropriate system stability criterions, which enhance the system stability and improve power quality, the two control loops of the PFC boost converter can be designed, as discussed in the following subsections.

### 4.2. Design of the Outer Voltage Loop

The design of the outer voltage control loop starts with obtaining the transfer function (TF) of the outer voltage circuit by modeling the small-signal stability of the boost converter, which describes the operation of the converter at the steady state operation point. For the boost converter shown in Figure 3,

$$\frac{V_o}{V_{in}} = \frac{1}{1 - D}, \tag{19}$$

$$1 - D = \frac{|v_s| \sin(\omega t)}{V_o} . \tag{20}$$

The boost converter inductor current ($I_{Lb}$) and diode current ($I_{Db}$) can be expressed as Equations (21) and (22), respectively.

$$I_{Lb} = |i_{Lb}| \sin(\omega t) \tag{21}$$

$$I_{Db} = I_{Lb} . (1 - D) \tag{22}$$

Substituting from (19)–(21) into (22), the complete boost converter diode current ($I_{Db}$) equation can be written as

$$I_{Db} = |i_{Lb}| \sin(\omega t) . \frac{|v_s| \sin(\omega t)}{V_o} = \frac{|i_{Lb}||v_s| \sin^2(\omega t)}{V_o} = \frac{1}{2} \frac{|i_{Lb}||v_s|}{V_o} - \frac{1}{2} \frac{|i_{Lb}||v_s|}{V_o} \cos(2\omega t), \tag{23}$$

where $\frac{1}{2} \frac{|i_{Lb}||v_s|}{V_o}$ refers to the DC component and $\frac{1}{2} \frac{|i_{Lb}||v_s|}{V_o} \cos(2\omega t)$ is refers to the AC component of the diode current.

Applying the averaged small signal perturbation to Equation (23) gives

$$i_{Db}^* = \frac{1}{2}\frac{|v_s| \cdot i_{Lb}^*}{V_o} + \frac{v_o^* \cdot I_{Db}}{V_o}. \tag{24}$$

The diode current is the sum of two components, load current ($I_{load}$) and the output capacitor current ($I_{Cb}$), which can be expressed as

$$I_{Db} = I_{load} + I_{Cb} = I_{cb} + \frac{V_o.}{R} \tag{25}$$

Applying the averaged small signal perturbation to Equation (25) with the resistive load case provides

$$i_{Db}^* = v_o^* C_b s + \frac{v_o^* \cdot I_{Db}}{V_o}. \tag{26}$$

Comparing the right-hand parts of Equations (24) and (26), we obtain

$$\frac{1}{2}\frac{|v_s| \cdot i_{Lb}^*}{V_o} = v_o^* C_b s. \tag{27}$$

So, the open loop TF of the outer voltage system is

$$G_v(s) = \frac{v_o^*(s)}{i_{Lb}^*(s)} = \frac{|v_s|}{2\,V_o\,C_b s}. \tag{28}$$

Figure 7 shows the control blocks for the outer voltage control loop of the PFC boost converter using the PI controller.

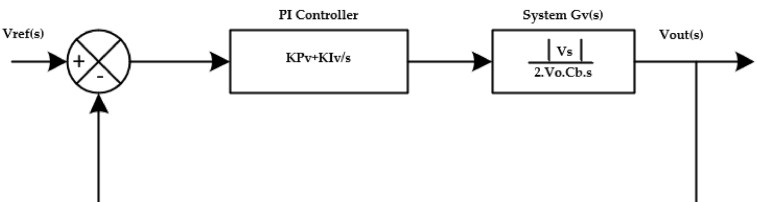

**Figure 7.** Block diagram of the outer voltage control loop using the PI controller.

The PI controller system can be described using the following TF:

$$G_{PIv}(s) = K_{Pv} + \frac{K_{Iv}}{s}, \tag{29}$$

where $K_{Pv}$ is the proportional gain and $K_{Iv}$ is the integral gain of the PI controller. The closed-loop TF of the outer voltage control loop system ($G_{CLV}(s)$) can be obtained as

$$G_{CLv}(s) = \frac{G_v(s) \cdot G_{PIv}(s)}{1 + G_v(s) \cdot G_{PIv}(s)}. \tag{30}$$

Substituting Equations (28) and (29) into (30) gives the closed-loop TF of the outer voltage control loop, as expressed by

$$G_{CLv}(s) = \frac{\frac{|v_s|}{2\,V_o\,C_b} \cdot (K_{Pv}\,s + K_{Iv})}{s^2 + \frac{|v_s|\,K_{Pv}}{2\,V_o\,C_b}s + \frac{|v_s|\,.K_{Iv}}{2\,V_o\,C_b}}. \tag{31}$$

So, the PI controller gains $K_{Pv}$ and $K_{Iv}$ can be designed using Equation (31) and the standard form of the second order system TF by selecting the optimal stability criteria for the control system bandwidth and the undamped natural frequency. Usually, the

bandwidth of the outer voltage loop must be very small to eliminate the harmonics of the DC bus voltage reflected by the AC input voltage at 60 Hz [8].

In this work, the closed loop bandwidth of the outer voltage loop ($W_n$) was assumed to be about 85 rad/s and the undamped natural frequency ($\xi$) was about 0.707. For the reliable operation of a controller with a wide loading range, the PI controller parameters were set to work with the minimum value of the load voltage ($V_o$) of 320 V and the rated input supply voltage ($V_s$) of 220 $V_{rms}$. Using these values, the closed-loop TF of the outer voltage control loop finally can be expressed as

$$G_{CLv}(s) = \frac{276.23\ K_{Pv}\ s + 276.23 K_{Iv}}{s^2 + 276.23\ K_{Pv}\ s + 276.23 K_{Iv}}. \tag{32}$$

Figure 8 shows the bode plot of the closed-loop TF of the outer voltage control loop, which shows that the controller offers unity gain for frequencies less than 28 Hz. This voltage control system working as a low-pass filter helps to remove the 60 Hz voltage ripple.

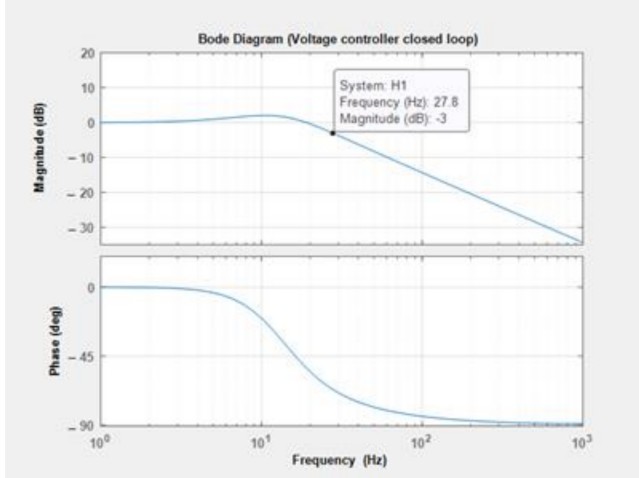

**Figure 8.** Bode plot of the closed loop outer voltage control system.

The optimal PI controller TF designed for the outer voltage control loop can be expressed as

$$G_{PIv}(s) = 0.435 + \frac{26.55}{s}. \tag{33}$$

To check the stability of the designed control voltage loop, the root locus of the closed-loop TF was plotted as shown in Figure 9, where the closed-loop eigenvalues are located in the negative side of the pole zero plane, so the closed-loop voltage system is inherently stable.

### 4.3. Design of the Inner Current-Control Loop

The design procedure for the inner current-control loop was the same as for the voltage control; we use an appropriate stability criterion for the inner current-control loop. The inner current-control loop must have enough bandwidth higher than the outer voltage loop bandwidth so that it is fast enough to track the current changes (to let the inductor current track the reference current). The current-control loop bandwidth must be less than switching frequency ($F_{sw}$) to reject the noise at the switching frequency. Using the small

signal model of the boost converter obtained in Equation (18), the open-loop TF of the inductor current control system $G_i(s)$ can be expressed as

$$G_i(s) = \frac{i_{Lb}^*(s)}{d^*(s)} = \frac{2V_o}{R(1-D)^2} \cdot \frac{1 + \frac{sRC_b}{2}}{1 + \frac{sL_b}{R(1-D)^2} + \frac{s^2L_bC_b}{(1-D)^2}}. \tag{34}$$

After rearranging Equation (34), the open loop TF of the inductor current system can be written as

$$G_i(s) = \frac{\left(\frac{V_o}{L_bC_bR}\right) \cdot (sRC_b + 2)}{s^2 + \frac{1}{C_bR}s + \frac{1}{L_bC_b}(1-D)^2}. \tag{35}$$

From Equation (35), we observe that the stability of the inductor current system depends on the converter duty cycle (D). In this work, the PFC boost converter was designed to obtain a constant output voltage with a wide range of the input voltage (85–265 V). Based on these operating conditions, the minimum duty cycle of the converter is about 0.40 and the maximum duty cycle is about 0.95.

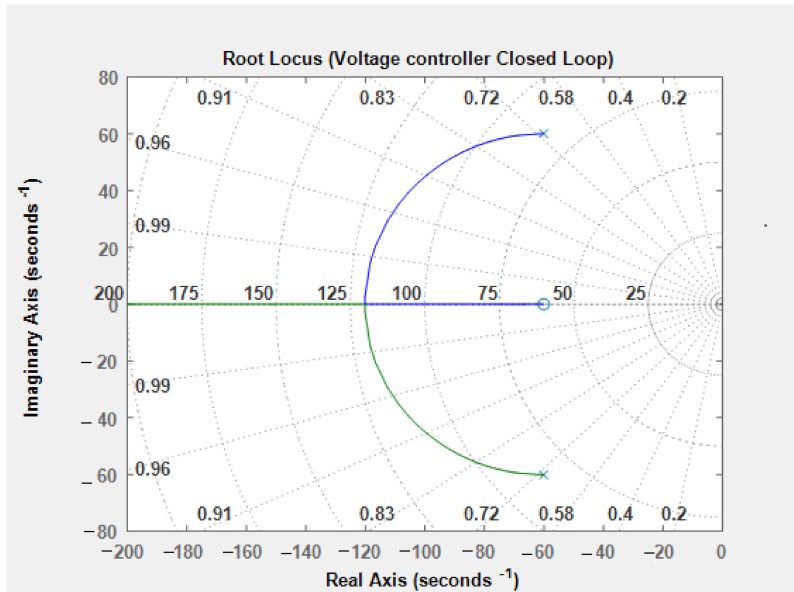

**Figure 9.** Root locus of the closed loop T.F of the outer voltage control system.

Before starting the design of the controller for the inner current-control loop, the inductor current system stability was checked using the root locus stability, which was plotted as shown in Figure 10 for the minimum and maximum converter duty cycles. At $D_{min} = 0.40$, the open loop roots $r_{1,2} = -6.97 \pm 826.82i$, and at $D_{max} = 0.95$, $r_{1,2} = -6.97 + 68.55i$. Varying the duty cycle from 0.4 to 0.95, the real part of the roots is usually negative; at constant value of about $-6.97$, only the imaginary parts change with the different values of the converter duty cycle, in which the current system of the CCM-PFC converter is absolutely stable in open loop with the operating range of the converter duty cycle (D).

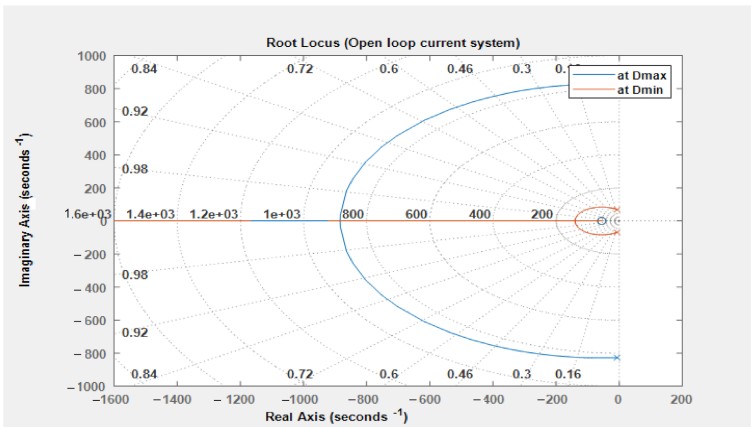

**Figure 10.** Root locus plot of the open loop current system with different duty cycle.

Figure 11 shows the bode plot of the open loop of the inductor current system. The peak resonance frequency is proportional to the duty cycle value, and the converter high bandwidth available is proportional to the high duty cycle of the converter.

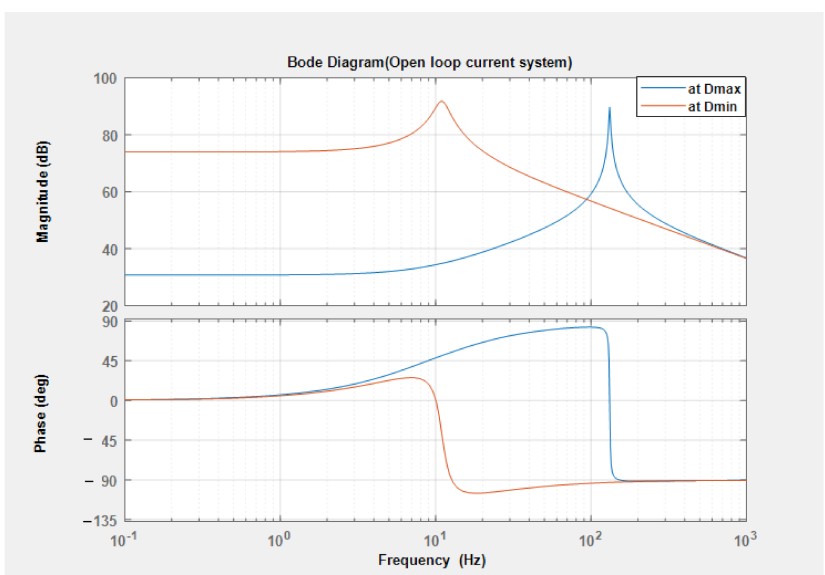

**Figure 11.** Bode plot of the open loop current system with different duty cycle.

Figure 12 depicts the block diagram of the inner current-control loop using the PI controller.

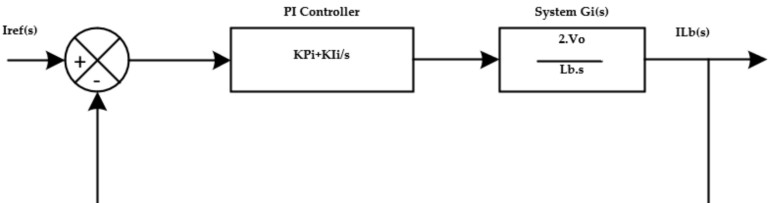

**Figure 12.** Block diagram of the inner current control system with PI controller.

The TF of the closed loop current-control loop can be expressed as

$$G_{CLI}(s) = \frac{I_{Lb}(s)}{I_{ref}(s)} = \frac{G_i(s) \, . G_{PIi}(s)}{1 + G_i(s) \, . G_{PIi}(s)} \; . \tag{36}$$

The TF of the PI current controller ($G_{PIi}(s)$) can be expressed as

$$G_{PIi}(s) = K_{Pi} + \frac{K_{Ii}}{s} \, . \tag{37}$$

where $K_{Pi}$ and $K_{Ii}$ are the proportional and integral gains for the PI current controller, respectively.

Since the converter switching frequency ($F_{sw}$) was selected as about 100 kHz in this work, for high frequency analysis, the capacitor can be shorted, and the open-loop TF of the inductor current system in Equation (35) can be simplified as

$$G_i(s) = \frac{2 \, V_o}{sL_b}. \tag{38}$$

Substituting Equations (37) and (38) into (36), the closed-loop TF for the inner current-control system can be derived as

$$G_{CLI}(s) = \frac{2 \, V_o \, K_{Pi} \, s + 2 \, V_o K_{Ii}}{s^2 L_b + 2 \, V_o \, K_{Pi} \, s + 2 \, V_o K_{Ii}}. \tag{39}$$

Using the same converter operating conditions of the voltage and power used in the voltage loop controller design and with a high bandwidth of about 5000 rad/s for the inner current-control loop, the closed-loop TF of the inner current-control loop can be expressed as

$$G_{CLI}(s) = \frac{(1.36 * 10^6) \, K_{Pi}s + (1.36 * 10^6) K_{Ii}}{s^2 + (1.36 * 10^6) K_{Pi} \, s + (1.36 * 10^6) K_{Ii}}. \tag{40}$$

Figure 13 shows the bode plot of the closed-loop TF of the current-control loop, which shows that the controller offers unity gain for frequencies less than 1610 Hz. This current control system, also working as a low pass filter, helps to remove the switching frequency noise.

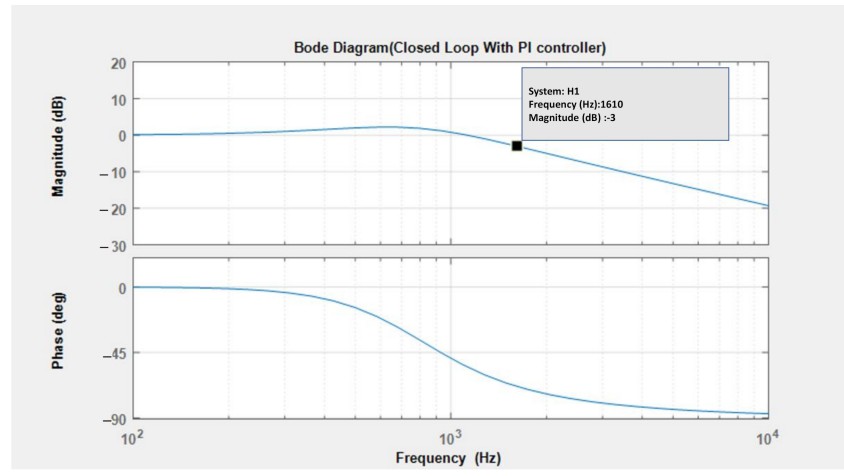

**Figure 13.** Bode plot of the current controller closed loop T.F with PI controller.

The optimal PI controller transfer function designed for the inner voltage control loop can be expressed as

$$G_{PIv}(s) = 0.005 + \frac{18.40}{s}. \tag{41}$$

Figure 14 shows that the Eigenvalues of the closed-loop current controller are located on the negative side of the pole zero plane, which ensures that the current control system has absolute stability.

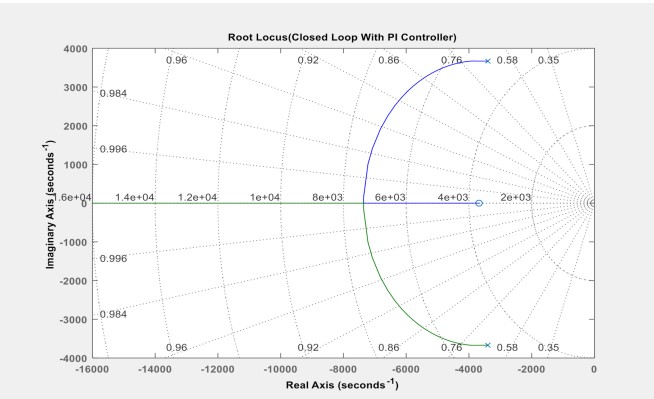

**Figure 14.** Root locus of the closed loop T.F of the inner current control loop.

The IP controller is characterized by a highly dynamic response and less overshoot compared with the PI controller. So, an optimal IP controller is proposed and designed in this section to replace the PI in the inner current-control loop to eliminate the current distortion around the zero-crossing point.

Figure 15 depicts the block diagram of the inner current control system using the IP controller consisting of two control loops instead of the one used by the PI controller, where the integral gain feeds in forward and the proportional gain feeds backward.

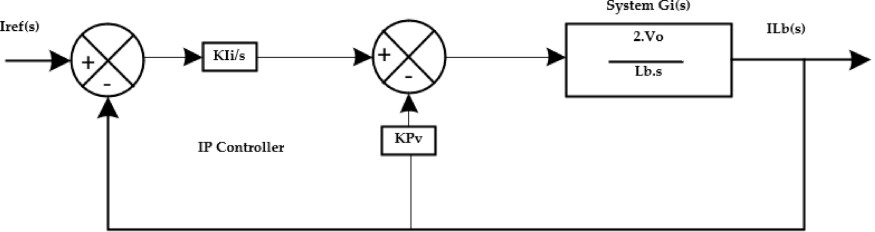

**Figure 15.** Block diagram of the inner current control system with the proposed IP controller.

From the block diagram in Figure 15, Equation (42) can be derived.

$$I_{Lb} = \left[ (I_{ref} - I_{Lb}) \frac{K_{Ii}}{s} - I_{Lb} K_{Pi} \right] \left[ \frac{2 V_o}{L_b s} \right]. \tag{42}$$

After rearranging Equation (42), we obtain

$$I_{Lb} \left[ \frac{2 V_o}{L_b s} \left( \frac{K_{Ii}}{s} + K_{Pi} \right) + 1 \right] = I_{ref} \left( \frac{2 V_o K_{Ii}}{L_b s^2} \right). \tag{43}$$

The closed-loop TF of the inner current-control loop using the IP controller can be expressed as

$$G_{CLI}(s) = \frac{I_{Lb}(s)}{I_{ref}(s)} = \frac{\frac{2 V_o K_{Ii}}{L_b s^2}}{\frac{2 V_o}{L_b s} \left( \frac{K_{Ii}}{s} + K_{Pi} \right) + 1} = \frac{2 V_o K_{Ii}}{s^2 L_b + 2 V_o K_{Pi} s + 2 V_o K_{Ii}}. \tag{44}$$

The problem of the slow response of the inductor current to tracking the reference current can be solved using the IP controller in the inner current- control loop. Since this controller structure provides two control loops for the current control system, not just the one used in the PI controller, the reliable and exact tracking of the inductor current to the reference current is obtained. As shown in Equation (44), there is no zero in the TF using the IP controller compared with the TF in Equation (39) with the PI controller, which also provides a system response with less overshoot.

To perform the comparison study between the PI and IP controllers based on the reliability of reducing the current distortion around the zero-crossing point, IP controller parameters were designed under the same operating condition and system stability criteria that were used in the design of the PI controller. Therefore, the closed-loop TF of the inner current-control loop using the IP controller can be expressed as

$$G_{CLI}(s) = \frac{(1.36 * 10^6) K_{Ii}}{s^2 + (1.36 * 10^6) K_{Pi} s + (1.36 * 10^6) K_{Ii}}. \tag{45}$$

Figure 16 shows the bode plot of the closed-loop TF of the current-control loop with the IP controller, which demonstrates that the controller offers unity gain for frequencies less than 823 Hz. This current control system with the IP controller also works as a low pass filter, which helps to remove the switching frequency noise. The Eigenvalues of the PI and IP controller are the same, which shows that the closed-loop TF of the current controller with the IP controller is also stable.

Figure 17 shows the bode plot of the closed-loop TF using the PI and IP controllers, demonstrating that the overshoot using the PI controller was eliminated using the IP controller. Figure 18 depicts the step response using both controllers; the overshoot of the current controller due to the input step response was reduced from 21.7% with the PI controller to 5.40% using the proposed IP controller.

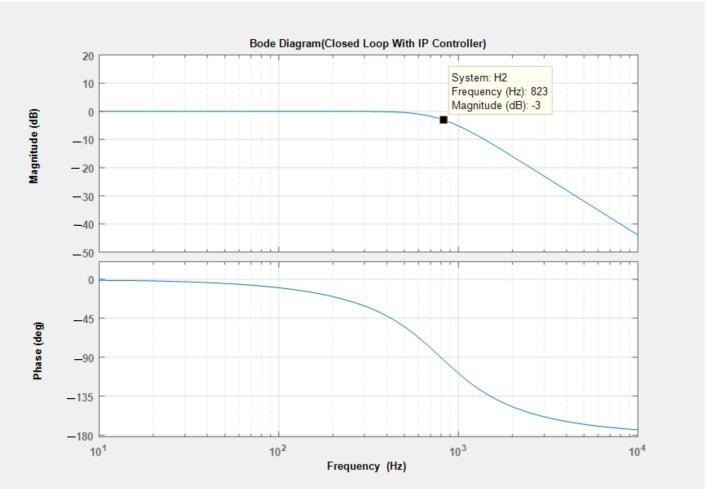

**Figure 16.** Bode plot of the current controller closed loop T.F with proposed IP controller.

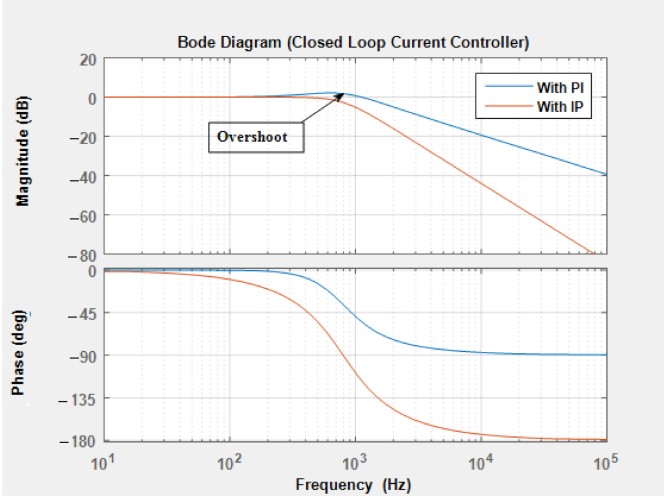

**Figure 17.** Bode plot of the inner current control loop with different controllers.

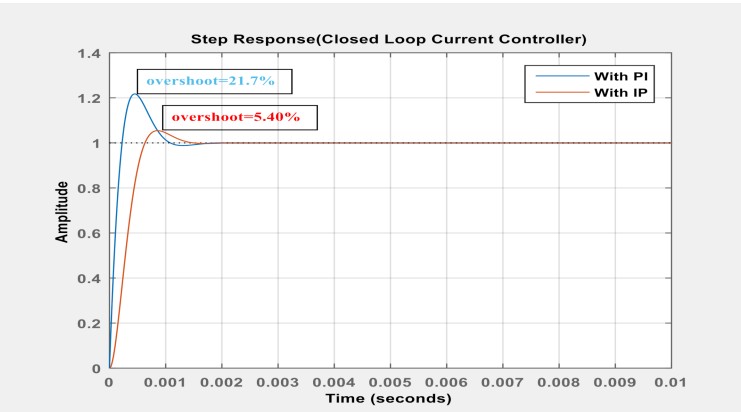

**Figure 18.** Step response of the inner current control loop with different controllers.

## 5. Design and Implementation of the Isolated Voltage Sensors

As mentioned in Section 1, one of the disadvantages of the digitally controlled PFC converter is that the voltage and current sensors that are required for interfacing with the MCU are expensive. In this section, for the economic and reliable operation of the designed digitally controlled PFC converter, two voltage sensors with low price components for sensing the AC input and the DC output voltage of the PFC converter are designed to interface with DSP TMS320F28335 (Texas Instruments, Dallas, TX, USA).

Figure 19 shows the schematic circuit of the proposed isolated DC voltage sensor. For DSP TMS320F28335 [31], the input voltage range of the analog digital converter (ADC) is 0 to 3 V DC voltage. So, the voltage sensors should be designed to convert the measured DC output voltage of the PFC converter to the available input range of the ADC of the DSP MCU.

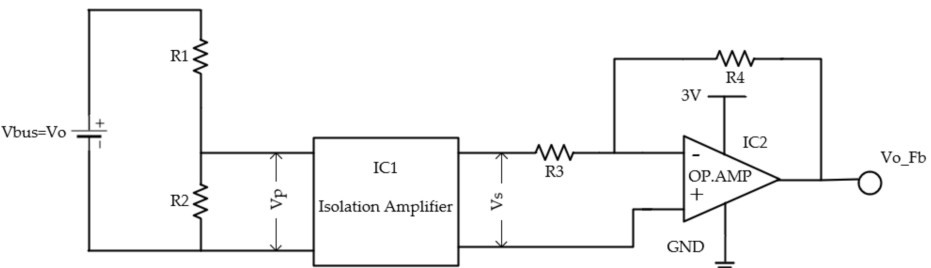

**Figure 19.** Block diagram of the designed isolated DC voltage sensor (V.S 2).

The voltage sensor consists of two stages. The first stage is implemented using the voltage dividers $R_1$ and $R_2$ and the isolation amplifier ($IC_1$) to offer isolation between the high-voltage power circuit and the low-voltage control circuit. The second stage is the operational amplifier ($IC_2$) circuit, which is used to adjust the sensor voltage to the required value based on its operation mode.

From the gain of the chosen isolation amplifier ($IC_1$) in the first stage, the values of resistors $R_1$ and $R_2$ can be designed as

$$V_{bus\_Max} = V_{P\_Max} \cdot \frac{R_2}{R_1 + R_2}. \tag{46}$$

where $V_{P\_Max}$ is the maximum primary voltage of the isolation amplifier and can be easily known from the datasheet of the chosen isolation amplifier (IC1). Then, by choosing the value of $R_1$, the value of $R_2$ can be calculated from Equation (46). An AMC1311 isolation amplifier from Texas instruments (TI) with a primary voltage range of about 2 V and unity voltage gain was used in the designed sensor, which provides galvanic isolation up to 7 kV,

low offset error, very low nonlinearity of less than 0.03%, and is inexpensive (about USD 3), which provides a highly reliable and economic design of isolated voltage sensing [32].

The inverting analog output from the AMC1311 IC is a fully differential signal and should provide bias voltage to the operational amplifier (IC2). The operational amplifier should be chosen to offer wide bandwidth and low slew rate time and low noise. The OPA320 operational amplifier from TI was used in this study, having a wide bandwidth of more than 20 MHz, a low slew rate time to the output voltage of about 10 V/µs, and costing only about USD 2.50 [33]. The expected total price of the proposed sensor is less than USD 10, which is very inexpensive compared with the other sensor types such as LV25-P sensor, which costs about USD 80, which was used for voltage sensing in a previous study [8].

The operational amplifier implemented in the proposed sensor works in inverting mode, so the close- loop voltage gain (β) can be expressed as

$$\beta = \left| \frac{V_{o\_Fb}}{V_{s\_Max}} \right| = \frac{R_4}{R_3}. \tag{47}$$

where $V_{s\_Max}$ is the maximum secondary voltage of the isolation amplifier and can be calculated using

$$V_{s\_Max} = V_{p\_Max} \cdot \alpha. \tag{48}$$

where $\alpha$ is the isolation amplifier voltage gain.

Choosing the $R_3$ value, the value of the resistance $R_4$ can be easily calculated using Equation (47).

Figure 20 shows the input output voltage curves for the designed isolated voltage sensor, demonstrating that for the input voltage of 400 V DC, the output voltage level is limited from 0 to 3 V with a very small slew rate time of about 25 µs.

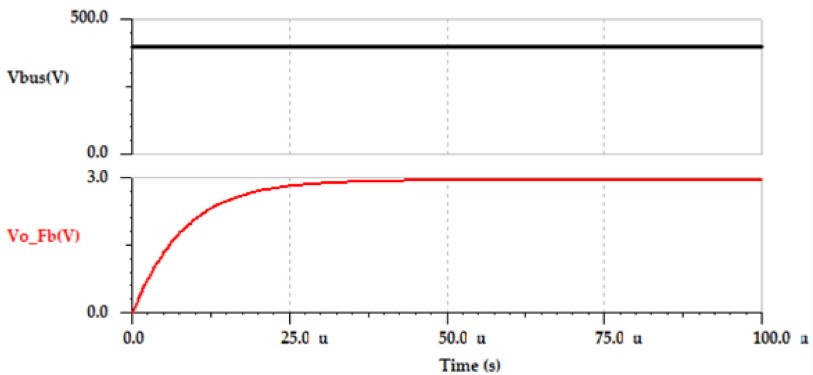

**Figure 20.** Isolated DC voltage sensor voltage curves with input voltage of 400 VDC.

Figure 21 shows the schematic circuit of the proposed isolated AC voltage sensor. Following the same steps for the DC voltage sensor design, the AC isolated voltage sensor was designed and the offset voltage ($V_{offset}$) was used to remove the negative voltage part from the sensed voltage to ensure the range of the DSP MCU input specification (0 to 3 V), as shown in the waveform of the input–output waveforms in Figure 22.

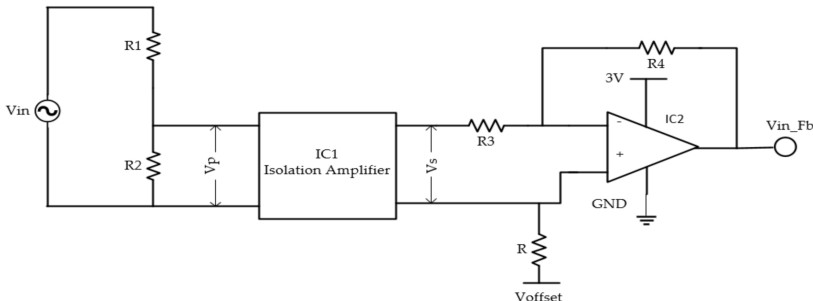

**Figure 21.** Block diagram of the designed isolated AC voltage sensor (V.S 1).

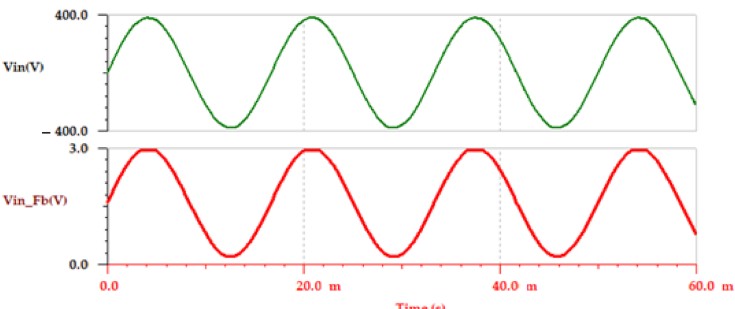

**Figure 22.** Isolated AC voltage sensor voltage curves with input voltage of 220 Vrms.

## 6. Simulation Results and Discussion

Figure 23 shows the complete schematic diagram of the CCM-PFC converter with the proposed digital control technique, which was simulated in PSIM software (Powersim, Rockville, MD, USA). The EMI filter was designed based on the design report of the 2.5 kW PFC analog converter from Infineon [29]. The digital control circuit of the PFC converter was implemented using DSP TMS320 F28335 with the DSP speed set to 150 MHz and the switching frequency of the pulse width modulation (PWM) generator to 100 kHz. In all sceneries of the current loop control, the voltage loop control was implemented using the PI controller.

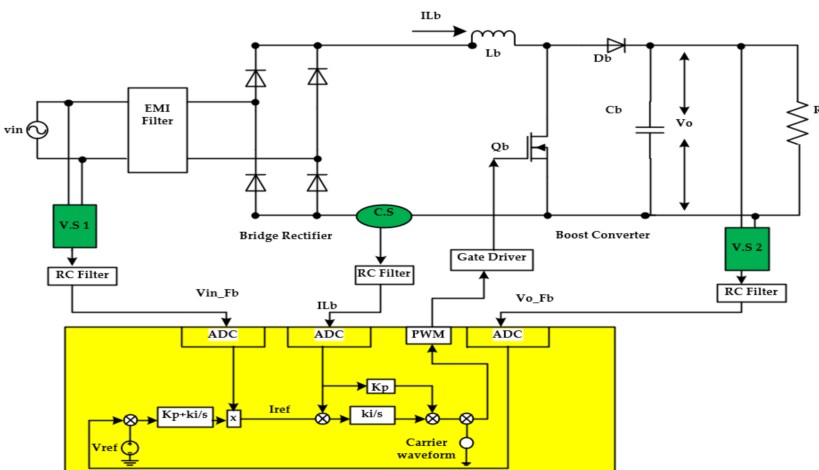

**Figure 23.** Complete schematic diagram of the CCM-PFC with the proposed digital control technique.

At the rated input of (220 $V_{rms}$, 60 Hz) and full load condition of $P_o$ = 2500 W and $V_{bus}$ = 400 V, Figure 24 shows the waveform of the inductor current ($I_{Lb}$) with the reference current (Iref) for the PI current controller. We observed distortion at the inductor current around the zero-crossing point due to the slow dynamic response of the PI controller and the failing of the inductor current ($I_{Lb}$) to track the reference current ($I_{ref}$).

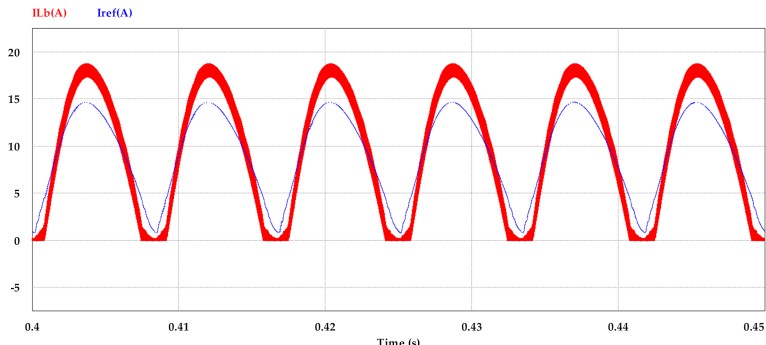

**Figure 24.** Waveforms of the inductor and reference currents with the PI current controller.

The distortion of the inductor current ($I_{Lb}$) at the zero-crossing point caused current distortion in the supply current ($I_{supply}$), as shown in Figure 25, from which we can observe the zero-crossing distortion (ZCD) period was about 1.45 ms at the supply current.

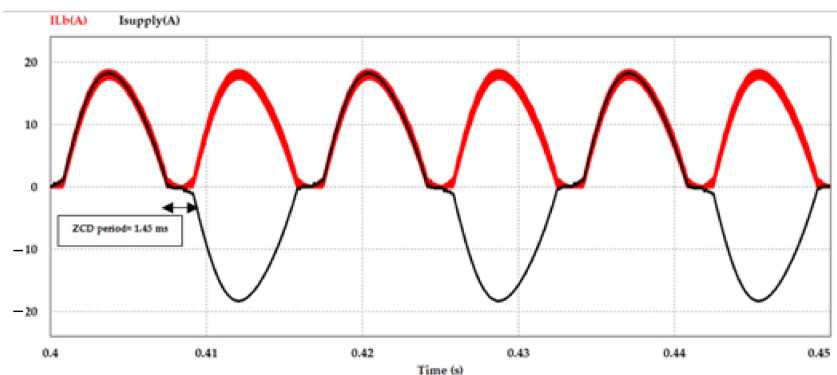

**Figure 25.** Waveforms of the inductor and supply currents with the PI current controller.

The ZCD produced the sinusoidal shape of the supply current; the current waveforms were unable to follow the sinusoidal waveform of the supply voltage ($V_{supply}$), as shown in the supply voltage–current waveforms in Figure 26.

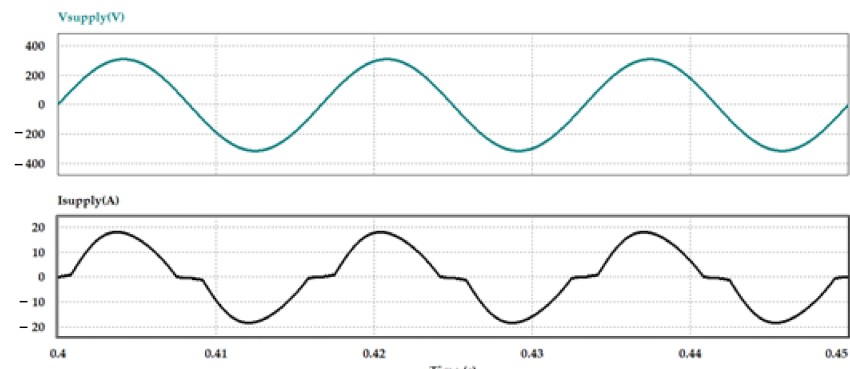

**Figure 26.** Waveforms of the supply current-voltage with the PI current controller.

Figure 27 depicts the frequency spectrum of the supply current. The fundamental current at 60 Hz was about 16.58 A, and the odd harmonics (3, 5, 7, . . . ) were the dominant harmonics present in the supply current frequency spectrum. The third harmonic component was about 1.81 A, the fifth harmonic was about 0.95 A, and the seventh harmonic was about 0.47 A. The THD in the supply current was 19.49% under the full load condition.

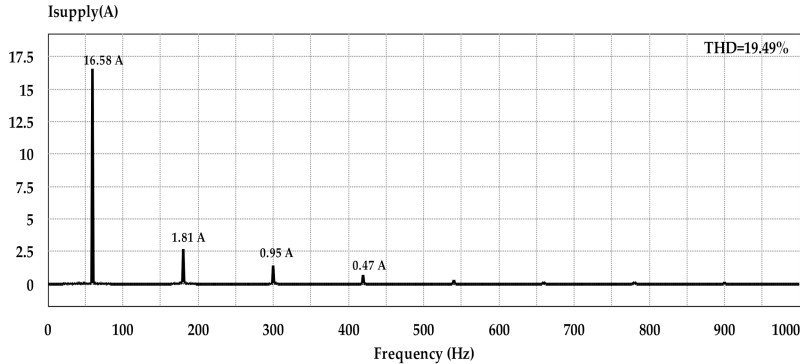

**Figure 27.** Frequency spectrum of the supply current at full load with the PI current controller.

At the rated input of (220 $V_{rms}$, 60 Hz) and full load condition of $P_o$ = 2500 W and $V_{bus}$ = 400 V, the IP digital controller was applied in the inner current-control loop to remove the distortion around the zero-crossing point. Figure 28 shows the waveform of the inductor current ($I_{Lb}$) with the reference current ($I_{ref}$) for the IP current controller. The inductor current ($I_{Lb}$) was successfully able to track the reference current ($I_{ref}$) and the ZCD significantly decreased.

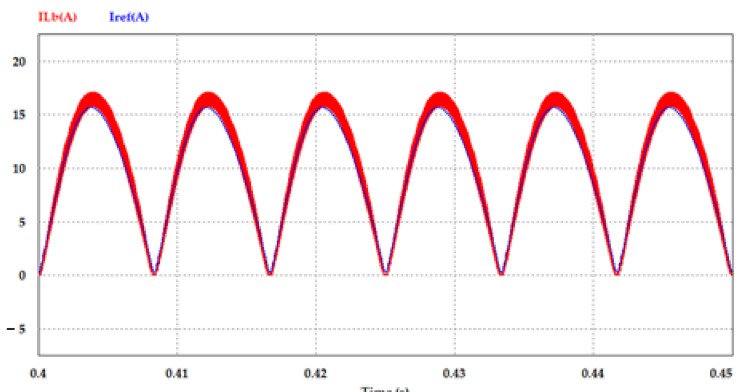

**Figure 28.** Waveforms of the inductor and reference currents with the proposed IP current controller.

Figure 29 shows the waveforms of the inductor and supply currents with the proposed IP current controller, from which we observed that the ZCD period decreased from 1.45 ms with the PI controller to about 0.17 ms using the proposed IP controller in the inner current-control loop of the PFC boost converter.

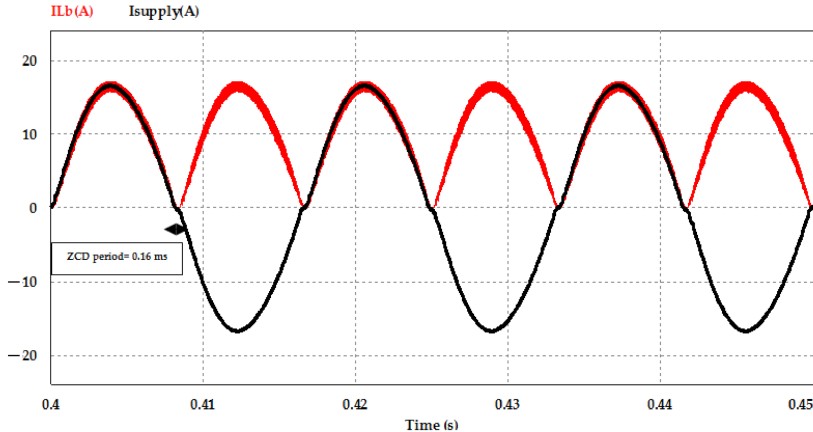

**Figure 29.** Waveforms of the inductor and supply currents with the proposed IP current controller.

Figure 30 shows the input supply voltage and current waveforms under the full load condition and using the proposed IP current controller. The reduction of the ZCD period made the current waveform approximately sinusoidal in shape and the voltage waveform was successfully followed.

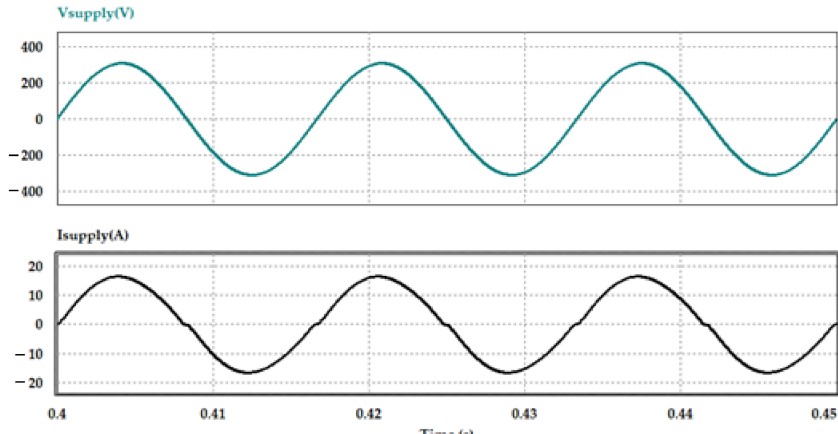

**Figure 30.** Waveforms of the supply current-voltage with the proposed IP current controller.

Figure 31 depicts the frequency spectrum of the supply current using the proposed IP current controller in the inner current-control loop. The fundamental current at 60 Hz was about 16.35 A. Using the IP current controller, the third harmonic component was reduced from 1.810 A to about 0.403 A, the fifth harmonic was reduced from 0.95 to 0.1855 A, and the seventh was reduced from 0.47 to 0.1133 A. The THD in the supply current under full load condition was reduced from 19.49% to about 5.23%.

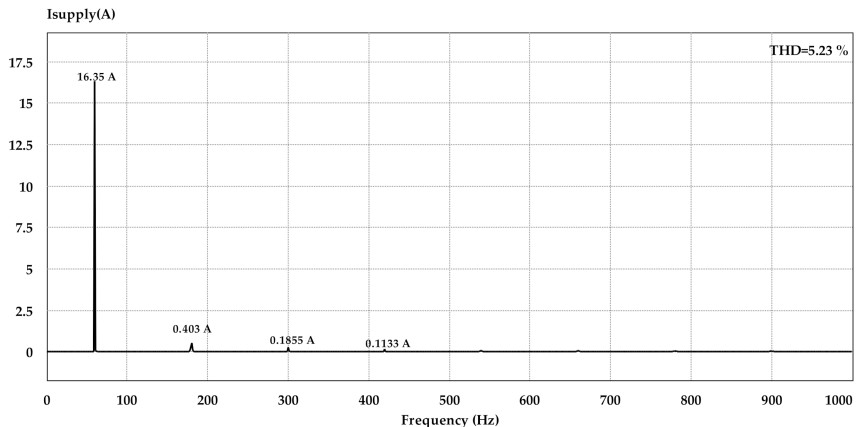

**Figure 31.** Frequency spectrum of the supply current at full load with the proposed IP current controller.

With different loading conditions, the simulation results of the designed PFC boost converter with different current control techniques was examined. Table 2 describes the PFC boost converter power factor and THD performance with loading conditions of 25%, 50%, 75%, and 100% of the full load condition (2500 W). From the results, under the full load condition, the designed PFC boost converter with the proposed current controlling technique solves the problem of the zero-current crossing distortion, reducing the THD to about 5.23% with input power factor about 99.93%. Under the light load condition of 25% loading condition, the THD value was about 34.8% with the PI controller and 30.20% with the IP controller. When the PFC converter operated at reduced load, discontinuous conduction mode (DCM) will appear during part of the line cycle or even the entire line which causes high distortion in the supply input current [34]. So, based on the THD and

power factor values, the optimum range of the designed converter loading condition is 75% to 100% of the full loading condition.

**Table 2.** Performance of the designed CCM-PFC with different loading conditions.

| Parameter | With Conventional PI Controller | | | | With Proposed IP Controller | | | |
|---|---|---|---|---|---|---|---|---|
| Load condition (W) | 625 | 1250 | 1875 | 2500 | 625 | 1250 | 1875 | 2500 |
| THD% | 34.80 | 26.50 | 21.39 | 19.37 | 30.20 | 20.70 | 11.00 | 5.23 |
| PF% | 93.86 | 96.38 | 97.52 | 98.31 | 94.50 | 96.60 | 99.12 | 99.93 |
| Power Losses (W) | 15.69 | 22.90 | 30.90 | 38.40 | 11.40 | 18.00 | 26.10 | 34.00 |
| Efficiency% | 97.48 | 98.17 | 98.35 | 98.47 | 98.18 | 98.57 | 98.13 | 98.66 |

## 7. Experimental Verification

Figure 32 shows the prototype of the designed digitally controlled PFC boost converter with the input EMI filter, boost converter, DSP control board, 20 A Hall current sensor and the designed isolated input output voltage sensors. The experiments were performed under the same operating conditions and circuit components values as those for the PSIM simulations shown in Table 1. The manufacturing description and part numbers for the components which used in the implementation of the boost converter are shown in Table 3.

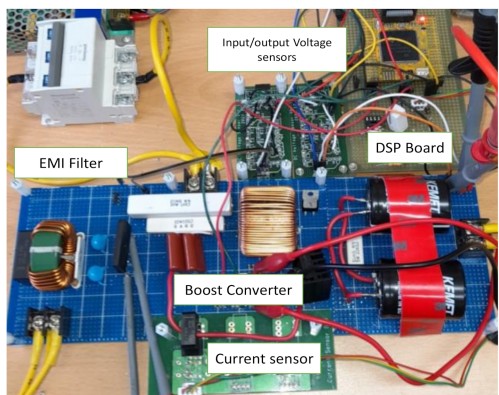

**Figure 32.** Prototype of the designed digitally controlled PFC boost converter.

**Table 3.** Boost converter component's specifications for the experimental setup.

| Component | Description/Part Number |
|---|---|
| Bridge Rectifier | GSIB2580 |
| PFC Inductor (Lb) | 2 stacked cores (Kool Mμ), 60 windings 1.15 mm copper wire (500 μH) |
| Switch (Qb) | IPW65R080 |
| Output Capacitor (Cb) | KEMET ALA7DA561DE450 (2 X560 μF) |

The photograph for the experimental setup of the PFC boost converter with the designed voltage sensors and DSP interface board using TMS320 F28335 DSP from Texas instruments is shown in Figure 33.

The designed converter performance was tested with using the maximum available load in our lab with power rating about 1000 W KIKUSI PLZ, Japan, 2019, DC electronic load at the rated input voltage of (220 V, 60 Hz). The DSP code for TMS320 F28335 was debugged and uploaded to the MCU using code composed studio (CCS), the PI controller was implemented in the voltage loop control with the different control techniques in the inner current control loop. The implemented voltage control loop able to control the output voltage to about 400 V with ripple voltage (peak to peak) of about 2.93% (11.70 V) which is less than the design specification of the output voltage ripple (20 V) as shown in the input-output voltage waveforms in Figure 34.

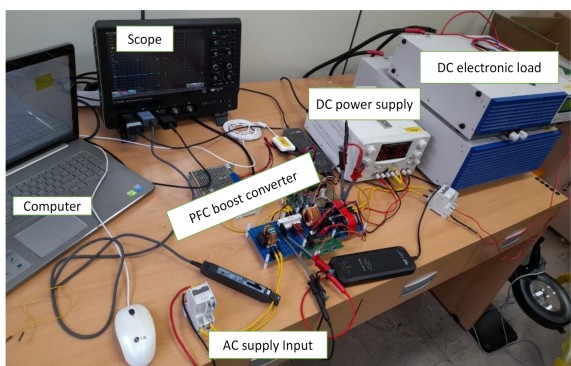

**Figure 33.** Experimental setup for testing of the designed digitally controlled PFC boost converter.

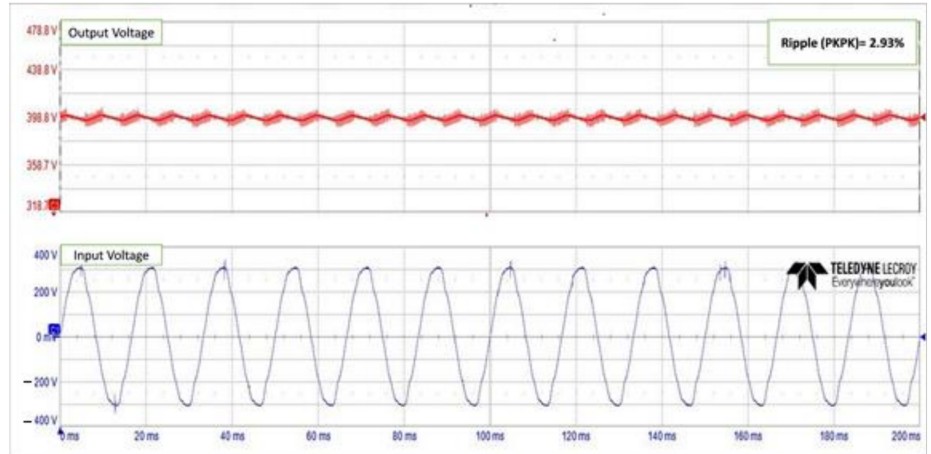

**Figure 34.** Input-output voltage waveforms of the designed PFC boost converter.

With the rated input voltage of (220 V, 60 Hz), and with using the different current control techniques (conventional PI and proposed IP) implemented in the inner current control loop, Figures 35 and 36 shows the experimental results for the input supply voltage-current, which we can notice that the proposed IP current controller with fast dynamic response able to reduce the zero-crossing distortion (ZCD) period and reduced the overall current distortion. The zero-crossing distortion period (ZCD) was reduced from 2.2 ms in case of the PI controller to about 1.6 ms with using the proposed IP current controller.

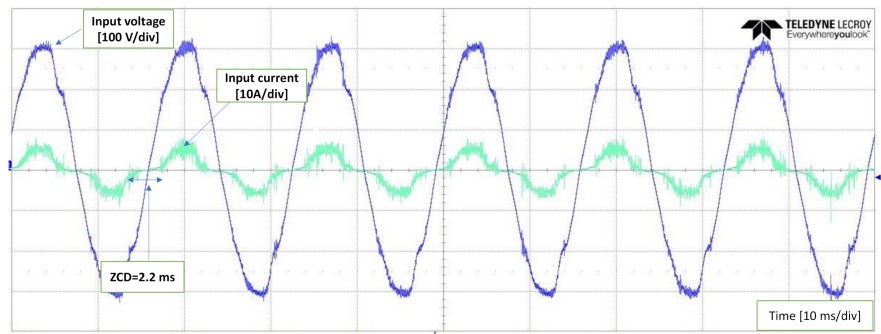

**Figure 35.** Experimental waveforms of the supply voltage-current with the conventional PI controller.

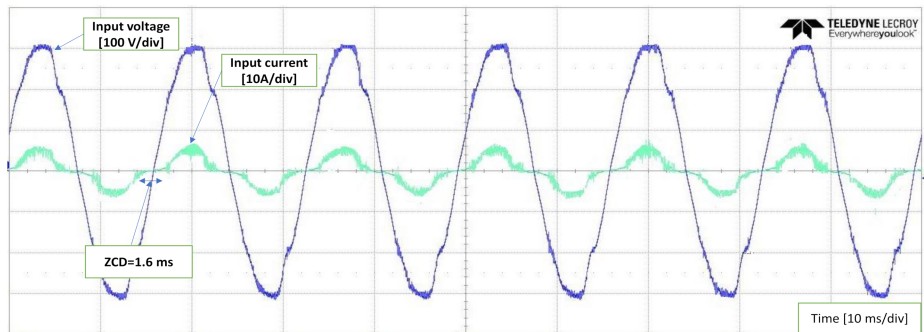

**Figure 36.** Experimental waveforms of the supply voltage-current with the proposed IP controller.

With using of the conventional PI current controller and with using load of 1000 W, the designed converter total power losses were about 33 W with efficiency about 96.70% and input power factor of about 96.10%. The implementation of the proposed IP controller in the inner current control loop reduced the total power losses to about 24.50 W, increased the efficiency to about 97.55% and the input power factor to about 97%. And, the highest budget (10.5 W) of these losses was accounted by the bridge rectifier internal resistance.

Figures 37 and 38 shows the experimental results for the input supply voltage and current with using the conventional PI and the proposed IP controllers respectively at 25% loading condition, which we can clear notice the zero-crossing distortion period (ZCD) of about 3.25 ms due to the slow dynamic response of the PI controller, and also notice the distortion in the current waveform due to operation mode transfer of the boost converter from CCM to DCM with light load condition [34]. The proposed IP controller reduced the ZCD period to about 3.05 ms but the current waveform still contains some distortions due to converter working in the DCM mode at light load condition.

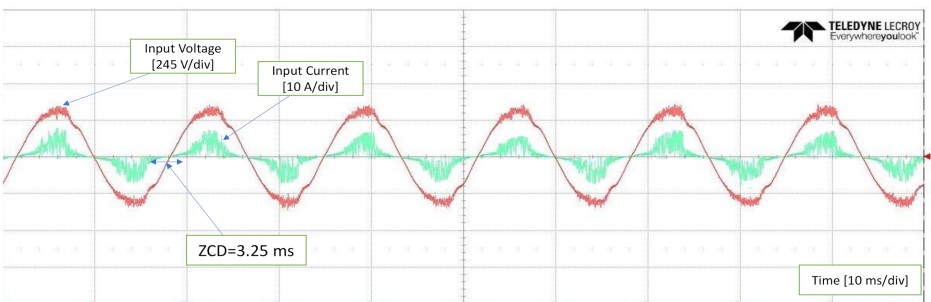

**Figure 37.** Experimental waveforms of the supply voltage-current with the conventional PI current controller at 25% loading condition.

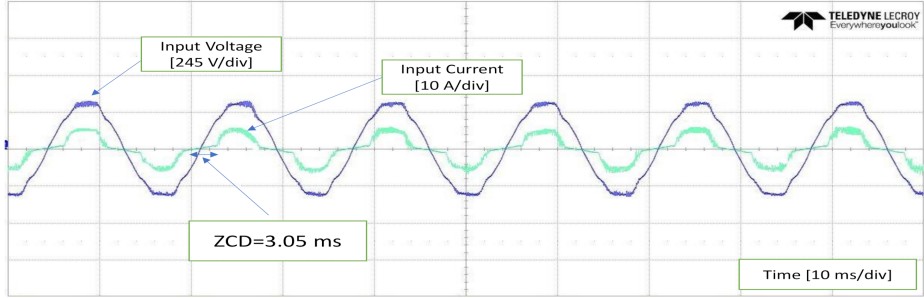

**Figure 38.** Experimental waveforms of the supply voltage-current with the proposed IP current controller at 25% loading condition.

The designed PFC boost converter based on the IP current controller in the inner current controller present lower THD with high efficiency, even higher than the conventional techniques based on the PI controller. However, in this paper for simplicity and since the parasitic resistance of the boost converter different components (Lb, Qb, Db, Cb) does not have more effects in the converter THD, the parasitic resistances were neglected in the modeling of the PFC boost converter in the simulation process. The effect of the parasitic resistance of the boost converter different components can be observes in the experimental results for the efficiency which is less than the simulation results for the same operating and loading condition. At 1000 W loading condition, the designed converter total power losses calculated using the simulation were about 16.5 W, but with the same loading condition the total power losses calculated with the experiment were about 24.5 W.

## 8. Conclusions

In this paper, a fast-dynamic response IP current controller was proposed based on small-signal stability modeling of the conventional PFC boost converter. The designed PFC converter controller's stability was verified using bode and root locus plots of the closed-loop control systems. Two low-cost voltage sensors for the AC input and the DC output measurements of the digitally controlled PFC converter were designed and experimentally used to interface with the DSP MCU TMS320 F28335. Complete prototype of the designed PFC boost converter with the DSP board and isolated voltage sensors was implemented in the lab. Experimental and simulation results show that the proposed IP current controller solves the problem of the current distortion around the zero-crossing point of the supply input current specially at high power density loading conditions, and reduces the THD to the standard value required by telecom power applications.

Comparative analysis of the conventional PI and the proposed IP current controllers was also performed, which showed that the inner current-control loop of the PFC converter with the proposed IP controller significantly reduced the zero-crossing distortion (ZCD) period with different loading conditions, reduced the THD to about 5.50% at full loading condition, and offers the PFC boost converter with input power factor more than 99%.

In future work and for a relatively accurate calculation for the efficiency and voltage regulation, the parasitic resistance of the boost converter different components (Lb, Qb, Db, Cb) can be considered in the modeling of the PFC boost converter to perform the impacts of the power losses and voltage drops due to parasitic resistance on the efficiency and voltage regulation of the designed PFC converter, which helps to increase the accuracy of the simulation results and make it closest to those which can be investigated in the experimental environment.

**Author Contributions:** A.H.O. created, designed and applied the proposed controlling technique. The literature review and manuscript preparation, as well as the simulations, were carried out by A.H.O. Experimental verification was carried out by A.H.O., H.J. and J.B. Final review of manuscript corrections was done by J.B. All authors have read and agreed to the published version of the manuscript.

**Funding:** This research was funded by grant(2018R1D1A3B0704376413) from National Research Foundation of Korea (NRF) and grant(20CTAP-C152903-02) from Ministry of Land, Infrastructure and Transport of Korean government.

**Data Availability Statement:** Not applicable.

**Conflicts of Interest:** The authors declare no conflict of interest.

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
