# Peer review of "Optimal IP Current Controller Design Based on Small Signal Stability for THD Reduction of a High-Power-Density PFC Boost Converter"

_applsci, doi:10.3390/app11020539_

Round 1
Reviewer 1 Report
In general, the work has good quality and a correct and extensive theoretical and simulation development, although it lacks experimental results (according to the authors, they will be obtained in a continuation of the work).
Aspects to correct:
- Correct the beginning of the line (45).
- Improve the quality of Figure 2.
- In this work, it is assumed that the Bosst converter always works in CCM mode, however the conditions are not detailed to guarantee this operating mode and that it does not change to DCM.
- In the detailed analysis in this work the parasitic resistance associated with L, D, Q and C have been ignored. It must be justified why they have been omitted and whether taking them into consideration would substantially modify the results of the work.
- The text indicates “P.F. more than 99% as required by almost of the telecom power application ”. This claim must be justified.
- The development of voltage sensors is trivial (voltage divider + isolator + inverter configuration with opamp). It is not something new, and as it is written in the abstract and conclusions, it seems an important contribution when it is not.
- I understand that the "Inverting analog output of the AMC1311" has been taken. If so, it should be clarified in the text.
- The conclusions should be expanded.
Author Response
Reviewer report #1
Dear professor Dr
First of all, we want to express our thanks and gratitude for the constructive comments and discussions to improve the quality of our manuscript significantly.
English was edited by using of the MDPI English editing service in order to increase the writing language quality of the manuscript, and the English edited version was used for the modification based on the reviewer’s comments.
In general, the work has good quality and a correct and extensive theoretical and simulation development, although it lacks experimental results (according to the authors, they will be obtained in a continuation of the work).
Above all, I appreciate for your thorough review on our paper. For the future work, as mentioned in our manuscript in the funding section, our work is funding from Korean government to design high power density telecom power supply for the 5G applications. And also, as mentioned in the manuscript introduction part, our design consists of two stages AC-DC PFC stage and the DC-DC output converter stage (The DC-DC converter stage was completely finished with simulation and experimental and published before in energies journal as reference [4] in our manuscript).
[4] Okilly, A.H.; Kim, N.; Baek, J. Inrush Current Control of High-Power Density DC–DC Converter. Energies 2020, 13, 4301.
The target of this manuscript is the modeling and control circuit designs for the AC-DC PFC stage but this take about 27 pages in the manuscript. So, we will extend this work in another future paper which will include the design and manufacturing of the PFC inductor, the design of the Printed circuit board of the PFC converter including the EMI filter, boost converter, DSP MCU control board and the voltage sensors for the experimental verification. Until now we already designed the PCB for the AC and DC voltage sensors as show in the following Figure 1 In this report. Also the DSP MCU controller circuit design using TMS320F28335, we just need to wait for receiving the required circuit elements and manufacturing of the PFC inductor for the boost converter, to complete the designed converter circuit for experimental verification.
Figure 1. PCB for the proposed AC and DC isolated voltage sensors in the future work.
I sincerely appreciate for your review again
Aspects to correct:
- Correct the beginning of the line (45).
Answer: Change was done, upper-case was used in the starting of the line as given in manuscript line:46.
- Improve the quality of Figure 2.
- Answer: Change was done, Figure have been edited with larger font sizes and high quality based on the manuscript line:121.
- In this work, it is assumed that the Boost converter always works in CCM mode, however the conditions are not detailed to guarantee this operating mode and that it does not change to DCM.Usually the two main reasons for the high total harmonic distortions in the PFC converters are:
- Answer:
- The slow dynamic response for the current controller.
- The mode transfer from CCM to DCM during the converter operation, specially when it is working with the light loading condition. as given in reference [35] added in our manuscript.
[35] De Gusseme K, Van de Sype DM, Van den Bossche AP, Melkebeek JA. Input-current distortion of CCM boost PFC converters operated in DCM. IEEE Transactions on Industrial Electronics. 2007 Mar 12;54(2):858-65.
- in our manuscript. The target converter designed to work in CCM with the IP current controller which solved the problem of the zero-crossing distortion as shown in the results of the THD in table 2 in our manuscript. At 25% of the load, we can notice that when the PFC converter operated at reduced load, discontinuous conduction mode (DCM) will appear during part of the line cycle and the THD still at higher (34.80% with PI controller, 30.20% with IP controller), this problem is common problem of the PFC converters at light loading condition [35].
- The optimum operation of our designed converter is with a high-power density where the converter controller enhanced the PFC operation at the CCM with power factor more than 99.90% and the THD value is about 5.23% (our project target to design power supply with power density more than 2 Kw).
thanks a lot professor for this constructive comment, and the following changes were added to our manuscript to describe this point:
- Discussions of the table 2 in the manuscript was expanded to include this comment in our manuscript lines 572 to 576.
- Also, the conclusion was expanded for clear explanation of the optimum loading range of the designed PFC converter for working in CCM operation with low THD and power factor more than 99%.
- In the detailed analysis in this work the parasitic resistance associated with L, D, Q and C have been ignored. It must be justified why they have been omitted and whether taking them into consideration would substantially modify the results of the work.[1] Bhat SX, Nagaraja HN. Effect of parasitic elements on the performance of buck-boost converter for PV systems. International Journal of Electrical and Computer Engineering. 2014 Dec 1;4(6):831.
- [2] Mohammad, Nur, Muhammad Quamruzzaman, Mohammad Rubaiyat Tanvir Hossain, and Mohammad Rafiqul Alam. "Parasitic effects on the performance of dc-dc sepic in photovoltaic maximum power point tracking applications." (2013).
- Answer: Since the main target of this manuscript is to design current controller to replace the slow dynamic response PI controller in the inner current control loop of the PFC boost converter to reduce the THD in the zero crossing point, and since the parasitic resistance doesn’t have impacts in the THD value of the converter (parasitic resistance impacts the converter efficiency due to the power losses and the output voltage regulation due to voltage drop (I as given in the references [1,2] cited in this report, and for the simplest mathematical analysis and design of the control circuit for the PFC converter, in small signal stability modeling the parasitic resistance of the boost converter had been neglected for the present work.
- This constructive review is very important comment for our future work, thanks a lot professor for this review again. Since the parasitic resistance of the boost converter impacts the efficiency and the voltage regulation of the designer converter, So, for the experimental part of this work, we can get the value of the parasitic resistances for different components after finish the manufacturing. Our converter modeling equations will modify to include the parasitic resistances of L, D, Q and C and comparative analysis with and without also can be provided based on the converter efficiency and voltage regulation.
- The conclusion was expanded to add the future work about this point in the manuscript lines:583-600
- The text indicates “P.F. more than 99% as required by almost of the telecom power application”. This claim must be justified.our project was funded from Korean government to design 5G telecom power supply that can offer the requirements of the Korean telecom companies such as LG, Samsung, SKT telecom. In Korean telecom companies, the DONGAH ELECOMM power supply is the most used type and the power factor design specifications of this power supply more than 99% as shown in the electrical design specifications in Figure 2 In this report. Also, for the international telecom companies such as Huawei telecom, design specifications of the power supply required that the power factor must be more than 99% as also shown in the Figure 3 In this report
- TEXAS instruments, INFINEON are the most worldwide famous companies to design the power supplies for the industrial applications such as Telecom, electrical vichels and UPS, the following two references is the examples from the two companies for the design of the AC-DC power supply for the telecom applications.
- Answer: Thanks a lot professor for this review. The sentence in the manuscript line 163-166 has been reformulated as follows (which usually required in the designed specifications of the telecom power applications)
[1] (>95% Efficiency, 1-kW analog control AC/DC reference design for 5G telecom rectifier) design report, Texas instruments. https://www.ti.com/lit/pdf/tiduet4.
[2] (PFC boost converter design guide - Infineon Technologies) design report, Infineon technology, https://www.infineon.com/dgdl/InfineonApplicationNote_PFCCCMBoostConverterDesignGuide-AN-v02_00-EN.pdf?fileId=5546d4624a56eed8014a62c75a923b05.
Figure 2. DONGAH ELECOMM AC-DC power supply electrical specifications.
Figure 3. Huawei telecom AC-DC power supply electrical specifications.
- The development of voltage sensors is trivial (voltage divider + isolator + inverter configuration with op-amp). It is not something new, and as it is written in the abstract and conclusions, it seems an important contribution when it is not.Thanks a lot professor for this review, in the abstract and conclusion parts, the section about voltage sensors was modified (where the verb (designed) used instead of the (proposed)). In order to express our work about the voltage sensors based on your review.Components of the voltage sensors is not new but our design can be reliable for using for the economic design of the complete power supply (the designed voltage sensor expected price not more than 10 $) since we need to design the complete AC-DC power supply from A to Z with price not more than 250 $, normally the price of the 2 Kw Telecom power supply is from 500 to 800 $ from international companies such as Texas or Infineon companies.
- As we mentioned in the introduction part, one of the major disadvantages of the PFC converter design based on the digitally MCU is the expensive prices of the voltage sensors required for the input and output voltage measurements reaches to about 80 $, our target from the design is to reduce the price of these voltage sensors for the economic design of the PFC digitally controlled stage, because also we have to compare between the analog and digital controlling technique based on the price and operation performance
- Answer:
- I understand that the "Inverting analog output of the AMC1311" has been taken. If so, it should be clarified in the text.
- Answer: Yes, the inverting analog output from the AMC1311 IC is a fully differential signal and should designed to provide bias voltage to the operational amplifier which also designed to work in the inverting mode.
Manuscript lines 476 and 478 were added to clarify the text about this point.
- The conclusions should be expanded. I sincerely appreciate for your review again
- Answer: The conclusion was expanded in the manuscript lines from 583 to 600.

Reviewer 2 Report
The paper needs to be edited to correct the English. In its current form it is very difficult to follow it through. Experimental results will significantly help with the quality of the paper.
Author Response
Reviewer report #2
The paper needs to be edited to correct the English. In its current form it is very difficult to follow it through. Experimental results will significantly help with the quality of the paper.
Dear professor Dr
First of all, we are apologizing for the English quality in the manuscript since the authors of the manuscript are not native English speakers. Also, we want to express our thanks and gratitude for any constructive comments and discussions to improve the quality of our manuscript significantly.
- English was edited by using the MDPI English editing service to increase the writing language quality of the manuscript (the following is the certificate from the MDPI English editing team.). The English edited version was used for the modification based on the reviewer’s comments.
- For future work, as mentioned in our manuscript in the funding section, our work is funding from the Korean government, to design a high-power density telecom power supply for the 5G applications. And also, as mentioned in the manuscript introduction part, our design consists of two-stage, the AC-DC PFC stage and the DC-DC output converter stage (the second stage was finished with simulation and experimental and published before in energies journal as the reference [4] in our manuscript).[4] Okilly, A.H.; Kim, N.; Baek, J. Inrush Current Control of High-Power Density DC–DC Converter. Energies 2020, 13, 4301.
- The target of this manuscript is the AC-DC PFC stage modeling and control circuit designs, this takes about 27 pages in the manuscript. So, we will extend this work in another future paper which will include the design and manufacturing of the PFC inductor, the design of the Printed circuit board of the PFC converter including the EMI filter, boost converter, DSP MCU control board, and the voltage sensors for the experimental verification.
- Until now we already designed the PCB for the AC and DC voltage sensors as shown in the following Figure 1 In this report and the DSP MCU circuit using TMS320F28335, we just need to wait for receiving the required circuit elements and manufacturing of the PFC inductor for the boost converter, to implement the complete designed converter circuit for experimental verification.
- Thanks a lot, professor again for the review of our manuscript and I hope that I have conveyed to you the idea that we have a lot of future work associated with this project, which we cannot put all in the same manuscript, also, our future research related to this research point was enclosed in the conclusion of the manuscript.
Figure 1. PCB for the proposed AC and DC isolated voltage sensors in the future work.
- I sincerely appreciate for your review again.
